# FEDERATED LEARNING WITH GAN-BASED DATA SYN-THESIS FOR NON-IID CLIENTS

## ABSTRACT

Federated learning (FL) has recently emerged as a popular privacy-preserving collaborative learning paradigm. However, it suffers from the non-IID (independent and identically distributed) data among clients. In this paper, we propose a novel framework, namely Synthetic Data Aided Federated Learning (SDA-FL), to resolve the non-IID issue by sharing differentially private synthetic data. Specifically, each client pretrains a local generative adversarial network (GAN) to generate synthetic data, which are uploaded to the parameter server (PS) to construct a global shared synthetic dataset. The PS is responsible for generating and updating high-quality labels for the global dataset via pseudo labeling with a confident threshold before each global aggregation. A combination of the local private dataset and labeled synthetic dataset leads to nearly identical data distributions among clients, which improves the consistency among local models and benefits the global aggregation. To ensure privacy, the local GANs are trained with differential privacy by adding artificial noise to the local model gradients before being uploaded to the PS. Extensive experiments evidence that the proposed framework outperforms the baseline methods by a large margin in several benchmark datasets under both the supervised and semi-supervised settings.

## 1 INTRODUCTION

The recent development of deep learning technologies has led to major breakthroughs in various domains. This results in a tremendous amount of valuable data that can facilitate the training of deep learning models for intelligent applications. A traditional approach to exploit these distributed data samples is to upload them to a centralized server for model training. However, directly offloading data raises severe privacy concerns as data collected by mobile clients may contain sensitive information.

By decoupling model training from the need of transferring private data to the cloud, federated learning (FL) offers a promising approach to collaboratively learn a global model without directly sharing the local data. Particularly, McMahan et al. (2017) introduced the Federated Averaging (FedAvg) algorithm that synchronously aggregates the updates of local models, while satisfying the basic requirements for privacy protection. Later, FedProx was proposed by Li et al. (2020) that adds a proximal term in the loss function to improve the statistical stability of the training process.

A key characteristic of federated learning is the non-IID (independent and identically distributed) data distribution among clients. Consequently, different clients learn from different data distributions, which leads to high heterogeneity among local models and degrades the effectiveness of model aggregation (McMahan et al., 2017; Wang et al., 2020b). For extreme cases with highly skewed data distribution, traditional federated learning methods even lack a theoretical guarantee of convergence (Yu et al., 2020; Rothchild et al., 2020). Recent studies have attempted to address the non-IID issue by constructing a global shared dataset based on the local data to mitigate the data imbalance effect (Krizhevsky et al., 2009; Zhao et al., 2018; Yoshida et al., 2020; Oh et al., 2020). Particularly, several methods (Krizhevsky et al., 2009; Yoshida et al., 2020) proposed data sharing strategies to distribute a percentage of raw data among clients, but naive data sharing seriously violate the privacy requirement in FL. Besides, data augmentation approaches (Zhang et al., 2021; Oh et al., 2020) generate synthetic samples without directly sharing private data. However, these methods are still susceptible to data leakage without a privacy-preserving mechanism. Thus, it is important to investigate how

to generate privacy-preserving synthetic samples, as well as how to effectively utilize the private dataset and synthetic dataset in the FL framework.

To alleviate the heterogeneity of the data distribution in FL without compromising privacy, we propose a novel framework, named Synthetic Data Aided Federated Learning (SDA-FL), which resolves the non-IID issue by sharing differentially private synthetic data. Under this framework, each client pretrains a local generative adversarial network (GAN) (Goodfellow et al., 2020) to generate synthetic data, thus avoiding sharing the raw data directly. A parameter server (PS) performs pseudo-labeling for the uploaded synthetic data and constructs a global shared dataset. In each global aggregation, the PS updates the pseudo labels of the synthetic dataset based on the received models and updates the global model with the confident synthetic data. This iterative labeling process constitutes a synthetic dataset with high-confidence labels, which is fed back to the clients to construct their training samples, including the local private data and synthetic data, following an almost identical distribution. Meanwhile, to provide a privacy guarantee for the generative models, we adopt the widely used differential privacy (Dwork, 2008) to ensure that the actual training samples can not be revealed by adversaries. In particular, we use the Wasserstein GAN with gradient penalty (WGAN-GP) (Gulrajani et al., 2017) as the backbone and introduce the privacy budget by adding artificial noise to the gradients in the training process. The SDA-FL framework is mostly compatible with many existing FL methods and can be applied in both supervised and semi-supervised settings.

We evaluate the proposed framework on several standard datasets under both supervised learning and semi-supervised learning, which shows its effectiveness in resolving the non-IID distribution problem even in the extreme case (with only one class of real data at each client). We also investigate the tradeoff between the privacy budget and the FL performance, which demonstrates that this framework is fully functional under the strict differential privacy protection requirement. Our major contributions are summarized as follows:

- We develop a global shared synthetic dataset by the differentially private GAN models. This data augmentation method effectively resolves the non-IID problem in FL and maintains the local data privacy.

- To utilize the global shared synthetic dataset efficiently, we propose a novel framework, SDA-FL, in which the PS provides and keeps updating high-confidence pseudo labels for the synthetic dataset iteratively, thus improving the performance of the global model.

- We evaluate our framework on several benchmark datasets under the high heterogeneous data distribution, where our framework consistently outperforms other baselines under both supervised and semi-supervised learning. Besides, our framework experiences a slight performance degradation with a stringent privacy budget.

## 2 RELATED WORK

**Non-IID Challenges in Federated Learning** The pioneering work in FL is FedAvg (McMahan et al., 2017), which simply averages model parameters trained by different clients in an element-wise manner, weighted proportionately by the clients' data size. However, this algorithm often leads to slow convergence and performance degradation when data is heterogeneous across local clients (Li et al., 2019). To alleviate this issue, Wang et al. (2020a) leveraged the reinforcement learning to intelligently choose the client devices to participate in each round of federated learning to counter-balance the bias introduced by non-IID data. Besides, some studies proposed personalized layers to adjust the local model according to the private dataset (Arivazhagan et al., 2019; Liang et al., 2020). For example, FedPer was introduced in Arivazhagan et al. (2019) that captures personalization aspects in federated learning by only using the local data to train the personalized layers. Knowledge distillation (Deng et al., 2020; Lin et al., 2020) is also a promising idea of personalized federated learning, which transfers information from other clients to a certain client. Although the aforementioned methods outperform FedAvg on strongly non-IID data, the extra computational costs are not affordable by the resource-constrained IoT devices (Zhu et al., 2021). Thus, client clustering has been proposed to construct a multi-center framework by grouping the clients into different clusters based on the model similarity (Ghosh et al., 2020; Kopparapu & Lin, 2020). As clients avoid the negative knowledge transfer from the dissimilar models (Pan & Yang, 2009), clustering methods can alleviate the negative effect of the non-IID data distribution. However, such methods are not scalable as the communication cost increases largely with the number of clients.

**Data Augmentation and Privacy Preserving** Recently, methods based on some form of data sharing have received increasing attentions and achieved prominent performance in FL (Zhu et al., 2021; Zhao et al., 2018). Particularly, data augmentation approaches showed the potential capability to address the non-IID issue by modifying the local distributions without sharing private data (Duan et al., 2019; Zhang et al., 2018; Eaton-Rosen et al., 2020). A vanilla data augmentation method was proposed by Duan et al. (2019), in which each client performs local data augmentation to reach a balanced global distribution. Besides, several methods (Oh et al., 2020; Shin et al., 2020; Yoon et al., 2021) leveraged the Mixup technique (Zhang et al., 2018) to conduct data augmentation, where the clients share the blended local data and collaboratively construct a new global dataset to tackle the non-IID issue. However, frequent data exchange may be susceptible to privacy issues. Recently, the GAN-based data augmentation framework (Bowles et al., 2018; Yoshida et al., 2020) was shown to be promising for reducing the degree of local data imbalance in FL. The general idea is to train a good generative model in the presence of non-IID data. Under this framework, each client sends its local seed samples to the server for the GAN training, and then a well-trained generator will be sent to all the clients for model updating. However, directly sending local data samples to the server violates the data privacy requirement. FedDPGAN (Zhang et al., 2021) suggested an alternative that all the clients train a global GAN together based on the FL framework to supplement the scarce local Covid-19 data. Unfortunately, the white-box generators sent to the server are vulnerable to adversarial attacks (Chen et al., 2020).

## 3 PRELIMINARY

**Federated Learning** FedAvg (McMahan et al., 2017) is one of the representative training algorithms in federated learning, and it trains a shared neural network without disclosing the raw data of clients. For every step $t = 0, \ldots, T-1$, every client $k \in \mathbb{S}_t$ downloads the global model $\boldsymbol{w}_t$ and updates the local model with local dataset $\mathbb{D}_k$ via stochastic gradient descent (SGD), i.e., $\boldsymbol{w}_{t+1}^k \leftarrow \boldsymbol{w}_t^k - \eta_{t+1}\nabla\ell$. The updated local models are then sent back to the PS for a weighted aggregation $\boldsymbol{w}_{t+1} \leftarrow \frac{1}{K}\sum_{k \in \mathbb{S}_t} \boldsymbol{w}_{t+1}^k$. These procedures repeat until convergence or all the $T$ training steps are exhausted.

**Differentially Private Generative Adversarial network** We utilize a differentially private generative adversarial network (DPGAN) for data augmentation while protecting local data privacy. To avoid the gradients vanishing and mode collapse problems (Arjovsky & Bottou, 2017; Arjovsky et al., 2017) that are commonly encountered by many GAN models (Mirza & Osindero, 2014; Radford et al., 2015; Odena et al., 2017), we adopt the WGAN-GP (Gulrajani et al., 2017), which is upgraded based on the Wasserstein GAN (WGAN) (Arjovsky & Bottou, 2017; Arjovsky et al., 2017), as the backbone of the generative model for a stable training procedure. Besides, we add noise to the gradient in the generator training process, which provides privacy control in terms of differential privacy (Torkzadehmahani et al., 2019). The definition of differential privacy is as follows:

**Definition 1.** (Differential privacy[1]): A random mechanism $\mathcal{A}_p$ satisfies ($\epsilon$,$\delta$)-differential privacy if for any output's subset ($\mathcal{S}$) and for any two adjacent datasets $\mathcal{D}$, $\mathcal{D}'$, the following probability inequality holds:

$$\mathbb{P}\left(\mathcal{A}_p(\mathcal{D}) \in \mathcal{S}\right) \leq e^\epsilon \cdot \mathbb{P}\left(\mathcal{A}_p\left(\mathcal{D}'\right) \in \mathcal{S}\right) + \delta. \tag{1}$$

Particularly, $\delta$ is an optimal probability that guarantees the validity of the inequality, and $\epsilon$ is the privacy budget indicating the privacy level, i.e., a smaller value of $\epsilon$ implies stronger privacy protection. To satisfy the ($\epsilon, \delta$)-differential privacy, we follow the work (Xie et al., 2018) and add the Gaussian noise to the gradients updated in each discriminator training iteration. The relationship between the noise variance and differential privacy is shown as follows:

$$\sigma_n = 2q\sqrt{n_d \log\left(\frac{1}{\delta}\right)}/\epsilon, \tag{2}$$

where $q$ and $n_d$ denote the sample probability and the number of discriminator training iterations in each training round, respectively.

---

[1]Although our work leverages the differential privacy to address the extra privacy problem introduced by the GAN-based data augmentation, the potential privacy risk of FL is still an open problem, which is out of the scope of our study.

**Pseudo Labeling** Although GAN-based data augmentation can generate synthetic data, the labels of the synthetic data are unknown, which hinders the training of the model through back-propagation. In semi-supervised learning, pseudo labeling is a widely used method in classification tasks to generate confident pseudo labels (Sohn et al., 2020). A hard pseudo label will be applied to the unlabeled data if the maximum class probability (MCP) of the inference result (Corbière et al., 2019) exceeds a predefined threshold $\tau$. In our experiments, the threshold is set to be 0.95.

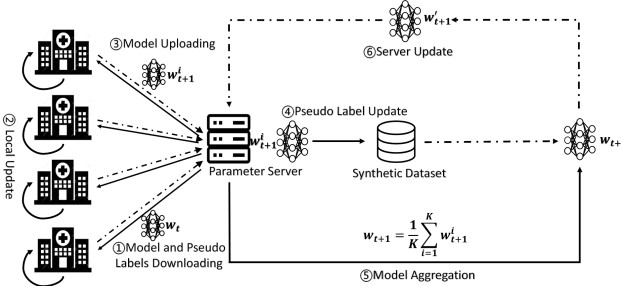

Figure 1: Overview of the proposed SDA-FL framework. Before the federated learning, the synthetic data from all clients are sent to the PS to construct a global synthetic dataset. In each iteration, every client first downloads the global model and updates the pseudo labels of the synthetic data for local training. The local models are then uploaded to the PS for pseudo label updating and model aggregation. Lastly, the PS updates the global model $\boldsymbol{w}_{t+1}$ with the updated synthetic dataset.

## 4 SYNTHETIC DATA AIDED FEDERATED LEARNING (SDA-FL)

We now introduce the SDA-FL framework that adopts GAN-based data augmentation to alleviate the negative effect of the non-IID data. The system diagram is shown in Fig. 1, and key algorithmic innovations built upon the classic FL framework are elaborated below.

**Global Synthetic Dataset Construction** At the beginning of the federated training, each client pretrains a local GAN model to generate synthetic samples based on its private data. Then, the synthetic samples are sent to the PS to construct a global shared synthetic dataset. To effectively leverage the synthetic dataset for federated learning, we perform pseudo labeling for these samples, which is critical for the effectiveness of the proposed framework.

In federated learning, the local model is the most suitable network to predict the pseudo labels for the synthetic data generated by the corresponding local GAN, as they are both trained with the same data. In our framework, to improve the confidence level of the pseudo labels, we utilize the local models to perform the pseudo labeling for the unlabeled synthetic data. Specifically, after receiving the local model $\boldsymbol{w}_{t+1}^i$ in each aggregation step, the PS utilizes the model predictions to assign a pseudo label for each unlabeled synthetic instance if the maximum class probability is higher than a predefined threshold $\tau$. In this way, we are gradually generating synthetic data samples with high-quality labels.

**Synthetic Data Aided Model Training** Augmented by the samples from the shared synthetic dataset that are with the confident labels, the data available for local training at different clients reach approximately homogeneous distribution. To make good use of the synthetic data, we leverage the $Mixup$ method (Zhang et al., 2018), which utilizes linear interpolation between the real sample $(x_i, y_i)$ and the synthetic sample $(\hat{x}_i, \hat{y}_i)$ to augment the real data:

$$\bar{x} = \lambda_1 \hat{x}_i + (1 - \lambda_1) x_i,$$
$$\bar{y} = \lambda_1 \hat{y}_i + (1 - \lambda_1) y_i,$$
(3)

where $\lambda_1$ follows the Beta distribution $\big(\mathrm{Beta}(\alpha, \alpha)\big)$ with $\alpha \in [0, 1]$. Combined with the cross-entropy loss, the Mixup loss for local update becomes:

$$\ell_1 = \lambda_1 \ell\big(f(\bar{\boldsymbol{X}}; \boldsymbol{w}_i), \hat{\boldsymbol{Y}}_i\big) + (1 - \lambda_1) \ell\big(f(\bar{\boldsymbol{X}}; \boldsymbol{w}_i), \boldsymbol{Y}_i\big).$$
(4)

**Algorithm 1:** Synthetic Data Augmented Federated Learning (SDA-FL)

---

**Input:** Local dataset $\mathbb{D}_k = \{\boldsymbol{X}_k, \boldsymbol{Y}_k\}$,
  Pretrained generators $\mathbb{G} = \{\boldsymbol{G}_k\}$
  $k \in \{1, \ldots, N\}$
Initialize $w_0$, Synthetic dataset $\hat{\mathbb{D}}_s = \{\hat{\boldsymbol{X}}_s, \hat{\boldsymbol{Y}}_s\}$ for server, $\hat{\mathbb{D}}_k = \{\hat{\boldsymbol{X}}_k, \hat{\boldsymbol{Y}}_k\}$ for clients
All clients upload the synthetic data generated by $G_k$ to PS
**for** $t = 0, \ldots, T - 1$ **do**
  $\mathbb{S}_t \leftarrow K$ clients are selected at random
  Send $\boldsymbol{w}_t$ and updated pseudo labels to clients $k \in \mathbb{S}_t$
  Clients Side:
  **for** $k \in \mathbb{S}_t$ **do**
    Updata local synthesis dataset$(\hat{\boldsymbol{X}}_k, \hat{\boldsymbol{Y}}_k)$
    $\boldsymbol{w}_{t+1}^k \leftarrow$
    $LocalUpdate(k, \boldsymbol{w}_t; \boldsymbol{X}_k, \boldsymbol{Y}_k, \hat{\boldsymbol{X}}_k, \hat{\boldsymbol{Y}}_k)$
  **end**
  Parameter Server Side:
  Pseudo Label Update
  $\boldsymbol{w}_{t+1} \leftarrow \frac{1}{K} \sum_{k \in \mathbb{S}_t} \boldsymbol{w}_{t+1}^k$
  $\boldsymbol{w}_{t+1} \leftarrow ServerUpdate(\boldsymbol{w}_{t+1}; \hat{\boldsymbol{X}}_s, \hat{\boldsymbol{Y}}_s)$
**end**

**Algorithm 2:** Model Update

---

$LocalUpdate(k, \boldsymbol{w}_t; \boldsymbol{X}_k, \boldsymbol{Y}_k, \hat{\boldsymbol{X}}_k, \hat{\boldsymbol{Y}}_k)$ :
$\boldsymbol{w} \leftarrow \boldsymbol{w}_t$
**for** $e = 0, \ldots, E - 1$ **do**
  Split $\mathbb{D}_k$ into batches of size $B$
  **for** $batch(\boldsymbol{X}, \boldsymbol{Y})$ **do**
    Sample batch$(\hat{\boldsymbol{X}}, \hat{\boldsymbol{Y}})$ from $(\hat{\boldsymbol{X}}_k, \hat{\boldsymbol{Y}}_k)$
    Use $(\hat{\boldsymbol{X}}, \hat{\boldsymbol{Y}})$ and $(\boldsymbol{X}, \boldsymbol{Y})$ to update $\boldsymbol{w}$ following (6)
  **end**
**end**
**return** $\boldsymbol{w}$
$ServerUpdate(\boldsymbol{w}_{t+1}; \hat{\boldsymbol{X}}_s, \hat{\boldsymbol{Y}}_s)$:
$\boldsymbol{w} \leftarrow \boldsymbol{w}_{t+1}$
**for** $e = 0, \ldots, E - 1$ **do**
  Sample batch$(\hat{\boldsymbol{X}}, \hat{\boldsymbol{Y}})$ from $(\hat{\boldsymbol{X}}_s, \hat{\boldsymbol{Y}}_s)$
  Evenly divide $(\hat{\boldsymbol{X}}, \hat{\boldsymbol{Y}})$ into $(\hat{\boldsymbol{X}}_1, \hat{\boldsymbol{Y}}_1)$ and $(\hat{\boldsymbol{X}}_2, \hat{\boldsymbol{Y}}_2)$
  Use $(\hat{\boldsymbol{X}}_1, \hat{\boldsymbol{Y}}_1)$ and $(\hat{\boldsymbol{X}}_2, \hat{\boldsymbol{Y}}_2)$ to update $\boldsymbol{w}$ following (6)
**end**
**return** $\boldsymbol{w}$

In addition, to keep more information of the local clients, another cross-entropy loss term is introduced for the real data as follows:

$$\ell_2 = \ell\big(f(\boldsymbol{X}_i; \boldsymbol{w}_i), \boldsymbol{Y}_i\big). \tag{5}$$

Then SGD is applied to update the local model:

$$\boldsymbol{w}_i \leftarrow \boldsymbol{w}_i - \eta_{t+1} \nabla(\ell_1 + \lambda_2 \ell_2), \tag{6}$$

where $\lambda_2$ is a hyperparameter to control the retention of the local information and will be set in different values in experiments. More details about the local update are shown in Algorithm 2.

In contrast to the traditional federated learning where the PS does not have access to any data to update the global model, the PS in our framework can keep the entire global synthetic dataset $\hat{\mathbb{D}}_s$ and uses it to update the global model, thereby helping to enhance the performance.

**Interplay between Model Training and Synthetic Dataset Updating** Because of the non-IID problem, there is a distinct difference in the weights of the local models among clients in federated learning. Furthermore, simply sharing the synthetic data with the PS and other clients will not result in the good training performance of the global model due to the lack of confident pseudo labels. Our framework, SDA-FL, deeply integrates federated learning and synthetic data, which effectively alleviates the non-IID issue and helps to train a good global model. As shown in Fig. 1, after creating the global synthetic dataset, it will be shared with all the clients so that the local data, including both the real and the synthetic data, approach an IID distribution. Without a doubt, the supplement of synthetic data results in the improvement of the local models. After being uploaded to the PS, the updated local models are used to perform the pseudo labeling and update the global synthetic dataset by the PS. The stronger local models can boost the confidence of the pseudo label. With the improvement of labeling for the global synthetic dataset, the PS can utilize it to update the global model and all the clients can improve their local models subsequently at the next-round training. Thus there is an interplay between the model update and the synthetic dataset at every training round: global synthetic dataset $\rightarrow$ global model update $\rightarrow$ local model update $\rightarrow$ labeling for global synthetic dataset, which results in a converged and well-perform model.

**SDA-FL vs. Traditional FL** From the above descriptions, the proposed SDA-FL framework introduces two additional operations at the server side, namely, pseudo label update of the shared synthetic dataset and server-side model update, as shown in Fig. 1. These two main innovations contribute to the performance improvement with non-IID data. In traditional FL algorithms (Li et al., 2020; Karimireddy et al., 2020), clients update their models based on local private data, which may lead to performance degradation when data are heterogeneous over clients. In our framework, the local dataset is augmented by the GAN-based synthetic samples, which alleviates the non-IID problem. Furthermore, the PS in the traditional FL system only performs model aggregation based on the uploaded local models. In contrast, our framework generates high-confidence synthetic data to update and improve the global model at the PS. Overall, with a shared synthetic dataset and an effective mechanism for pseudo label update, SDA-FL can overcome the heterogeneous data distributions among clients and enhance the model update at the server. We envision that this framework can be applied to develop other data augmentation-based methods for more efficient federated learning.

## 5 EXPERIMENTS

In this section, we evaluate the proposed SDA-FL framework in the presence of non-IID data for both supervised learning and semi-supervised learning. The experimental results on different benchmark datasets demonstrate the effectiveness of the proposed framework compared with baseline methods. Besides, ablation studies are conducted to illustrate the impacts of key factors on the generated samples and the model convergence.

### 5.1 EXPERIMENTAL SETUP

**Datasets** We implement the SDA-FL framework in the scenario where all the real data are stored in clients and the PS only has access to the synthetic data. We use four benchmark datasets, including MNIST (LeCun et al., 1998), FashionMNIST (Xiao et al., 2017), Cifar-10 (Krizhevsky, 2009), and SVHN (Netzer et al., 2011) to test the proposed method. The details about the datasets, models, and experimental settings are deferred to Appendix A. To demonstrate the effectiveness of our framework in handling different non-IID scenarios, we provide extensive experiments in the following five tasks.

**Tasks** **1) Federated supervised learning:** We compare our framework with FedAvg (McMahan et al., 2017), Fedprox (Li et al., 2020), Scaffold (Karimireddy et al., 2020), Naivemix, and Fedmix (Yoon et al., 2021) on MNIST, FashionMNIST, Cifar-10, and SVHN datasets.
**2) Federated semi-supervised learning:** We extend our framework to the semi-supervised learning setting in MNIST, FashionMNIST, and Cifar-10 datasets by performing pseudo labeling for the unlabeled data. To show the effectiveness of our SDA-FL method, we compare it with Local Fixmatch+Mixup (Diao et al., 2021), Local Fixmatch (Sohn et al., 2020), and Local Mixup (Zhang et al., 2018).
**3) The trade-off between the privacy budget and performance:** To investigate the influence of the differential privacy mechanism on the quality of the synthetic data and the model convergence, we validate our framework under different privacy budgets. In particular, we select the Fréchet Inception Distance (FID) (Heusel et al., 2017) as a metric to quantify the quality (similarity) of the synthetic data.
**4) The effectiveness of updating pseudo labels and the server update:** Compared with the traditional federated learning, SDA-FL leverages the pseudo labeling mechanism to generate high-confidence pseudo labels for the synthetic data, which are also used by the PS to further update the global model. To demonstrate the efficacy of these two mechanisms, we conduct ablation studies on a varying number of rounds for updating the pseudo labels and a varying number of server update steps.
**5) Ablation studies of the computational cost and the number of samples for training the generators:** To evaluate the performance of the generative models, we conduct ablation studies to investigate the influence of the training epochs and the number of samples on the quality (similarity) of the synthetic data.

Table 1: Test accuracy of different methods in various datasets. We report the best accuracy of Fedprox and Fedmix by tuning the parameters $\mu = 0.001, 0.01, 0.1, 1.0$ and $\lambda = 0.01, 0.1, 0.2, 0.5$, respectively.

| #class/client | 1 | | | | 2 | | | | 3 | | | |
|---|---|---|---|---|---|---|---|---|---|---|---|---|
| Algorithm | Mnist | FashionMnist | Cifar10 | SVHN | Mnist | FashionMnist | Cifar10 | SVHN | Mnist | FashionMnist | Cifar10 | SVHN |
| FedAvg | 83.44% | 16.50% | 18.36% | 14.05% | 97.61% | 73.50% | 61.28% | 81.11% | 98.42% | 82.47% | 79.33% | 84.18% |
| Fedprox | 84.17% | 57.14% | 11.24% | 17.53% | 97.55% | 75.76% | 63.16% | 86.28% | 98.38% | 83.43% | 79.54% | 92.15% |
| Scaffold | 25.39% | 56.80% | 12.81% | 11.64% | 94.17% | 70.82% | 60.78% | 73.34% | 96.89% | 77.68% | 79.35% | 80.13% |
| Naivemix | 84.35% | 66.62% | 14.39% | 14.35% | 84.35% | 79.54% | 64.39% | 84.64% | 98.11% | 82.09% | 78.92% | 92.30% |
| Fedmix | 90.96% | 72.11% | 13.57% | 16.78% | 90.96% | 82.41% | 65.76% | 86.61% | 98.46% | 84.65% | 79.49% | 92.61% |
| **SDA-FL** | **98.19%** | **85.70%** | **37.70%** | **88.46%** | **98.26%** | **86.87%** | **67.89%** | **90.70%** | **98.50%** | **87.06%** | **84.56%** | **93.16%** |

## 5.2 PERFORMANCE OF SDA-FL ON NON-IID FEDERATED LEARNING

**1) Federated Supervised Learning** We compare the performance of various methods under the same federated setting. With varying numbers of classes per client, the experimental results in Table 1 show that our framework outperforms other methods by a large margin, which is attributed to the GAN-based data augmentation that mitigates the detrimental effects of the data heterogeneity. Under the severe non-IID scenario (i.e., each client only has one class of data), our SDA-FL method maintains an accuracy of 88.46% in the SVHN classification task, while other baselines suffer from great performance degradation. In the Cifar-10 experiments, our framework outperforms the classic Naivemix and Fedmix algorithms at least by 5.0% with three classes of data at each client. This indicates that augmentation via synthetic data is more effective than averaging real data samples, although Fedmix adds another derivative term when averaging real images to explicitly handle the non-IID problem.

Recently, GANs have been widely used to solve the non-IID and data scarce issues based on the federated learning in the medical field (Nguyen et al., 2021; Zhang et al., 2021), which can afford the additional computational cost for training the GANs. To further show the feasibility of SDA-FL, we include another comparison on a Covid-19 dataset (Chowdhury et al., 2020). See more details about the Covid-19 experiment in Appendix B.

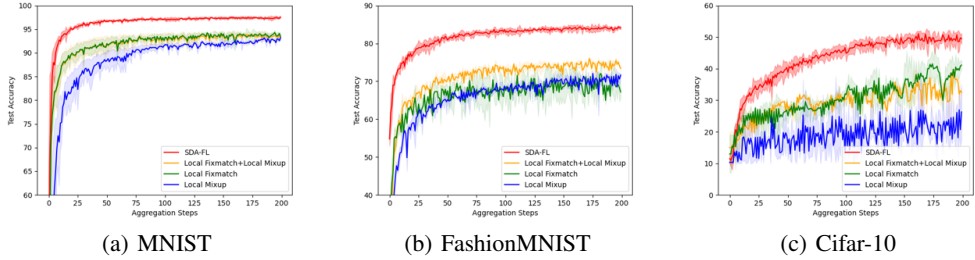

| (a) MNIST | (b) FashionMNIST | (c) Cifar-10 |
|---|---|---|

Figure 2: Test accuracy of different methods for federated semi-supervised learning on MNIST, FashionMNIST, and Cifar-10 classification tasks.

**2) Federated Semi-Supervised Learning** The results in Fig. 2 show that our framework achieves faster convergence and better performance than other algorithms, indicating its robustness and generalizability. Particularly, compared with Local Fixmatch+Local Mixup, our method improves the accuracy by nearly 10% in the FashionMNIST classification task. In the Cifar-10 dataset, the baseline methods are not able to train a usable global model (with a test accuracy below 40%), while our framework converges in this challenging scenario and improves the test accuracy by a significant margin. This is because the proposed pseudo labeling mechanism can provide high-quality labels for the synthetic and unlabeled samples, which benefits the federated training process.

**3) The trade-off between the privacy budget and performance** The privacy budget $\epsilon$ provides the privacy protection for the generation of the synthetic dataset, but a small value of $\epsilon$ affects the quality of the synthetic samples and thus degrades the training performance. To investigate the impact of the privacy budget, we evaluate the performance of the proposed method under different

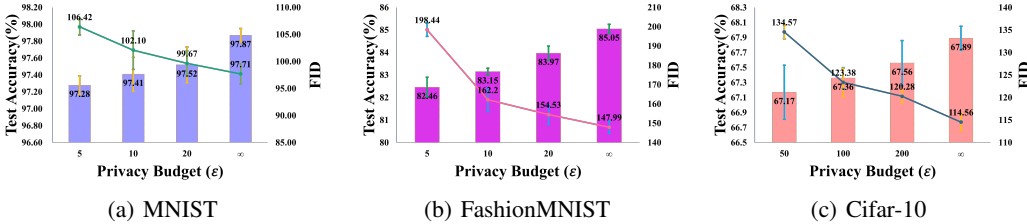

(a) MNIST       (b) FashionMNIST       (c) Cifar-10

Figure 3: Test accuracy and FID as a function of the privacy budget. We run three trails, and report the mean and the standard deviation of the test accuracy. The FIDs of the real samples in MNIST, FashionMNIST, and Cifar-10 are 10.54, 23.17, and 42.70, respectively, which are much larger than those of the synthetic data. It illustrates that the differentially private synthetic data, while of low image quality, still effectively help to improve the training performance with non-IID data.

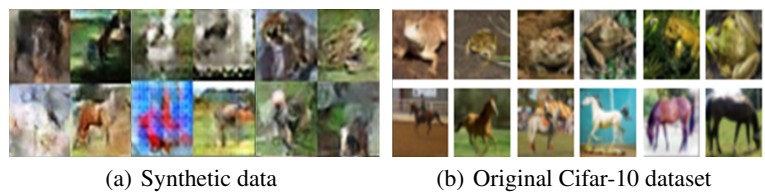

(a) Synthetic data       (b) Original Cifar-10 dataset

Figure 4: Visual comparison between synthetic data and the original Cifar-10 dataset. The synthetic data are sampled from the generator trained with privacy budget $\epsilon = 200$.

values of $\epsilon$. As shown in Fig.3, the proposed framework suffers from performance degradation with a small privacy budget. Compared with a protection-free scenario ($\epsilon = \infty$), setting a strict privacy budget ($\epsilon = 5$) (Xie et al., 2018) leads to around 0.61% and 2.59% accuracy drop in the MNIST and FashionMNIST datasets, respectively. Besides, we utilize the FID to measure the quality of the generated samples, where a smaller FID score denotes a better image quality. As illustrated in Fig.3, decreasing the privacy budget $\epsilon$ increases the FID score, which implies the degradation of the quality of the generated samples. Note that although our SDA-FL method is trained under the strict privacy requirement, compared with the results in Table 1, our framework still maintains desired performance over other methods.

In the Cifar-10 classification task, as the generators are sensitive to the Gaussian noise added to the gradient, we adopt a large privacy budget ($\epsilon = 50$) when training the proposed method. Fig.3(c) shows that our framework outperforms Fedmix and Fedprox in federated supervised learning. To illustrate the privacy guarantee on the generators in our framework, we show some synthetic samples in Fig.4(a), which are extracted from the generators trained with $\epsilon = 200$. In comparison with the samples from the private datasets, the synthetic images are blurry and noisy, which means the GAN models are not memorizing the real data, and the local privacy is not undermined. Besides, the performance improvements shown in Fig.3(c) and Table 1 indicate that the synthetic images with low perceptual quality can capture the necessary information for the model training.

**4) The effectiveness of updating pseudo labels and the server update** In comparison to the traditional federated learning, our framework can update the global network with the synthetic data at the server, which has the potential to further improve the performance. The results in Fig.5 show that the model performance on FashionMNIST will drop nearly 3% without any server update. However, too many server updates also degrade the performance because too much involvement of the synthetic data will relieve the effects of the local update. The empirical results show that the model trained solely with the synthetic data (the $\infty$ steps) can only get 66.0% accuracy, which highlights the necessity to effectively utilize the synthetic data as well as the local data for model training.

Besides, keeping updating pseudo labels for the synthetic data improves the model performance. As illustrated in Fig.6, the accuracy gradually increases when continuously updating the pseudo labels,

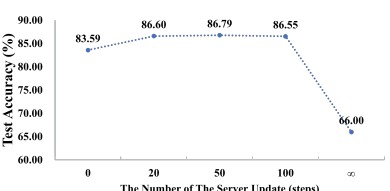

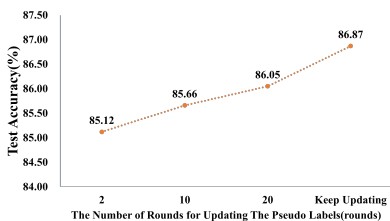

Figure 5: Test accuracy on FashionMNIST with varying server update steps. The $\infty$ steps mean that the model is only trained with the synthetic data.

Figure 6: Test accuracy on FashionMNIST with different rounds of pseudo label updating.

which demonstrates that our framework can improve the confidence level of the pseudo labels in the training process. Note that since our framework only updates the pseudo labels instead of the samples, the extra communication overhead is neglectable.

**5) The ablation studies of computational cost and the number of samples for training the generators** We present the results in Fig.7 and Fig.8, which show that insufficient training rounds and training samples for generators result in low-quality synthetic data and degraded performance. The generator only gets an FID score of 217.81 (23.17 for real data) and an accuracy of 83.76% when training the generators with 30 rounds. In addition, as shown in Fig.8, under the fixed 30-round computational cost, using fewer samples for training the generators reduces the quality of the synthetic data and thus affects the performance. Nonetheless, despite the low computational cost (30 rounds) and the small number of samples (1000 samples) used to train the generators individually, our framework still outperforms other baselines (82.41% in Fedmix), demonstrating the robustness of the proposed method.

In addition to the influence of the quality of synthetic data on the performance, other factors, such as the threshold $\tau$, $\lambda_2$, and the size of the synthetic dataset, also affect the global model. More ablation studies about these factors are shown in Appendix C.

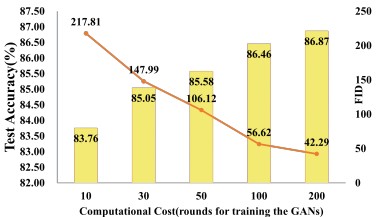

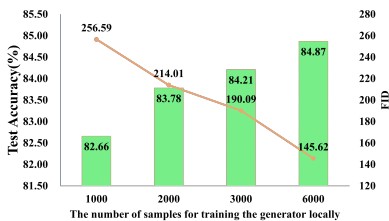

Figure 7: Test accuracy and FID on Fashion-MNIST, under varying training rounds for the GANs.

Figure 8: Test accuracy and FID on Fashion-MNIST, under varying samples for training the GANs locally.

## 6 CONCLUSIONS

We proposed a new data augmentation method to resolve the heterogeneous data distribution problem in federated learning, by introducing pretrained GANs to construct a differentially private global shared dataset. To effectively utilize the synthetic data, a novel framework, named Synthetic Data Aided Federated Learning (SDA-FL), was developed, which generates and updates confident pseudo labels for the synthetic data samples. By supplementing the non-IID private datasets with high-quality synthetic data samples, this framework effectively improves the consistency among local models. Experiment results showed that SDA-FL outperforms many existing baselines by remarkable margins in various image classification tasks and is still functional under strict differential privacy protection.

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

## A    EXPERIMENTAL DETAIL

**Datasets**    To demonstrate the generalizability of our framework, we use five widely used datasets to test our frameworks, where the details of the datasets are summarized in Table 2. In all the experiments, we equally divide the training samples and assign them to the clients based on the non-IID setting. Specifically, given the number of classes per client as $C$ and with $K$ clients, the whole dataset will be split into $K * C$ subsets, and each subset only has a single class. Then all the subsets of data will be randomly shuffled and distributed to clients. For example, in the MNIST experiment, under the scenario that each client only accesses two classes of data, given $K = 10$ clients, each

client keeps $60000/10 = 6000$ samples, each class with 3000 samples. The data distribution is kept the same in the comparison between our framework and other baselines. Except for the Covid-19 dataset, we set 10 clients for all the experiments, and all the clients participate in the training at every communication round.

**Models** For fair comparisons, we adopt a simple CNN model that consists of two convolution layers and two fully-connected layers for the MNIST and FashionMNIST classification tasks. Meanwhile, ResNet18 (He et al., 2016) is used for classifying the Cifar-10, SVHN, and Covid-19 image datasets. To generate the qualified synthetic samples, we set a generator with four deconvolution layers and a discriminator with four layers convolution layers followed by a fully-connected layer.

Table 2: The statistics of the datasets

| Dataset | MNIST | FashionMNIST | Cifar-10 | SVHN | Covid-19 |
|---|---|---|---|---|---|
| Training Samples | 60000 | 50000 | 50000 | 73257 | 12000 |
| Testing Samples | 10000 | 10000 | 10000 | 26032 | 3153 |
| Total dataset classes | 10 | 10 | 10 | 10 | 3 |

Table 3: The details of parameters in federated supervised learning and federated semi-supervised learning

| Parameters | Supervised | Semi-supervised |
|---|---|---|
| Learning rate ($\alpha$) | 0.03 | 0.03 |
| Batch size ($B$) | 64 | 16 (labeled data)+64 (Unlabeled data) |
| Local steps($E$) | 90 | SDA-FL: 35 (39 for Cifar-10), baselines: 40 |
| Server update steps | 50 (10 for Cifar-10) | 50 (0 for Cifar-10) |
| Communication rounds($T$) | 200 | 200 |
| Training rounds for GANs (90 steps/round) | 200 (400 for Cifar-10) | 200 (400 for Cifar-10) |
| Number of clients ($K$) | 10 | 10 |
| Fraction of clients | 1.0 | 1.0 |
| The size of the synthetic dataset | 40000 | 40000 |
| $\lambda_2$ (for our framework) | 1.0 (0 for Cifar-10) | 1.0 (0 for Cifar-10) |
| Threshold ($\tau$, for our framework) | 0.95 | 0.95 |
| $\mu$ (for Fedprox) | 0.001, 0.01, 0.1, 1.0 | / |
| $\lambda$ (for Naivemix and Fedmix) | 0.01, 0.1, 0.2, 0.5 | / |
| $M_k$ (for Naivemix and Fedmix) | 9 | / |

Table 4: The experimental settings of the ablation studies.

| Ablation study | Variable | Value |
|---|---|---|
| Computational cost | Training rounds for GANs | 10, 30, 50, 100, 200 |
| The samples for training GANs | The number of samples used for training the GANs locally | 1000, 2000, 3000, 6000 |
| The size of the synthetic dataset | The size of the synthetic dataset | 2000, 10000, 20000, 40000, 50000 |
| $\lambda_2$ | $\lambda_2$ | 0.25, 0.5, 0.75, 1.0, 2.0, 5.0 |
| Threshold ($\tau$) | Threshold ($\tau$) | 0.5, 0.7, 0.8, 0.95 |

**Experimental settings of tasks** **1) Federated supervised learning:** In the federated supervised and semi-supervised experiments, the parameters always are kept the same in our framework, and they are shown in the Table 3. In the baseline experiments, we set $\mu = 0.001, 0.01, 0.1, 1.0$ for Fedprox, and $\lambda = 0.01, 0.1, 0.2, 0.5$ for Naivemix as well as Fedmix. The best performances of them are reported in the Table 1.
**2) Federated semi-supervised learning:** In this experiment, we compare our framework with other baselines in MNIST, FashionMNIST, and Cifar-10 datasets. Every client only keeps two classes of data, containing 50 labeled instances, and the rest are unlabeled. For fair comparisons, the total gradients steps of SDA-FL are the same as other methods.
**3) The trade-off between the privacy budget and performance:** We evaluate SDA-FL in MNIST and FashionMNIST with the privacy budget $\epsilon = 5, 10, 20, \infty$ and 30 training rounds (each round with 90 steps) for GANs. In the Cifar-10 experiment, we set the privacy budget $\epsilon = 50, 100, 200, \infty$ to train the GANs with 400 training rounds (each round with 90 steps). Every client keeps two classes of data in all experiments.
**4) The effectiveness of updating pseudo labels and the server update:** In this task, we test our

framework in the FashionMNIST dataset and distribute two classes of data to every client. In our framework, the PS updates the pseudo labels for the synthetic dataset every communication step to maintain the confidence of the synthetic data. Compared with the federated supervised learning, we change to update the pseudo labels in the first $2, 10, 20$ communication steps instead of updating the pseudo labels at all the communication steps in our framework. In the ablation study of the server update, we perform the experiments with different steps of server update $(0, 20, 50, 100, \infty)$ and batch size $B = 64$.

**5) The ablation studies of computational cost and the number of samples for training the generators:** The settings of all the ablation studies are the same as the federated supervised learning, and we test them in FashionMNIST with 2 classes of data at each client. All the experimental settings of ablation studies are shown in Table 4.

Table 5: The data distribution of the Covid-19 experiment

| Hospital | 0 | 1 | 2 | 3 | 4 | 5 |
|---|---|---|---|---|---|---|
| Data | Normal 2000 Covid-19 750 | Covid-19 750 Pneumonia 250 | Pneumonia 250 Normal 2000 | Normal 2000 Covid-19 750 | Covid-19 750 Pneumonia 250 | Pneumonia 250 Normal 2000 |

Table 6: Test accuracy on Covid-19 dataset. The FedAvg (IID) represents the scenario where the training samples are distributed to all the clients by average based on the IID distribution.

| Algorithms | FedAvg | Fedprox | Scaffold | Naivemix | Fedmix | FedDPGAN | FedAvg (IID) | **SDA-FL** |
|---|---|---|---|---|---|---|---|---|
| Test Accuracy | 94.05% | 95.03% | 94.30% | 94.14% | 94.28% | 94.57% | 95.19% | **96.25%** |

# B THE APPLICATION IN THE COVID-19 DATASET

Because of the scarcity of Pneumonia data, we only set up six clients in this experiment, each of which keeps two classes of data, and the data distribution is shown in Table 5. We test SDA-FL with 30 local steps, 10 server steps, and 50 rounds for training the generators locally. In addition to the baselines in the federated supervised learning task, we also add FedDPGAN (Zhang et al., 2021) for comparison, which trains a global GAN to solve the non-IID issue for medical applications, and the results are shown in Table 6.

Our framework outperforms the other baseline algorithms by a remarkable margin. Particularly, SDA-FL is superior to FedDPGAN by 1.68% in the test accuracy, which indicates that the GANs which are trained individually generate higher-quality synthetic data than the global GAN trained based on the federated learning framework. Furthermore, in addition to addressing the non-IID issue, we also claim that the synthetic data can augment the data distribution, as SDA-FL is superior to FedAvg (IID) by 1.14% in the test accuracy. As a result, we assert that SDA-FL is able to show its advantages of solving non-IID problems and generating high-value data in medical applications.

# C ABLATION STUDIES

**The size of the synthetic dataset** Before the federated learning, all the clients constitute the synthetic dataset together. Larger synthetic datasets can typically provide more qualified data to clients and PS, thus improving performance, but they require larger storage space for clients. The results in Fig.9 show that, although the test accuracy on FashionMNIST decreases as the size of the synthetic dataset declines, our framework achieves 85.44% test accuracy with each client only uploading 200 samples, which is acceptable in exchange for less storage space requirement.

**The hyper-parameter $\lambda_2$** $\lambda_2$ is the parameter to control the use of the local real data. The results in Fig.10 show that compared with the normal settings of $\lambda_2$, the test accuracy on FashionMNIST drops by nearly 5.0% with $\lambda_2 = 0.25$, indicating that the training of the models is heavily reliant on the real data. Besides, an excessive $\lambda_2$ also degrades the performance since it relieves the effects of the synthetic data and cannot mitigate the non-IID problem as much as possible. In comparison, a moderate $\lambda_2$ is capable of combining the real data and the synthetic data intelligently and thus enhancing the models and solving the non-IID issue.

**Threshold** $\tau$    Threshold $\tau$ sets the criterion for selecting the high-confidence pseudo labels. The results on FashionMNIST are shown in Fig.11, which indicates that a small threshold impairs the performance because some synthetic data with low-confidence pseudo labels are still considered qualified and used to train the models. An excessive threshold, on the other hand, filters out a lot of qualified data, which affects performance. Therefore, $\tau = 0.95$ is a reasonable value to achieve a balance between quality and the quantity of the pseudo labels.

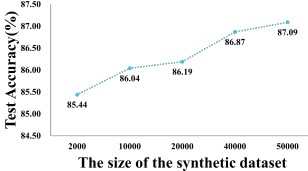

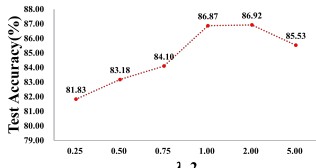

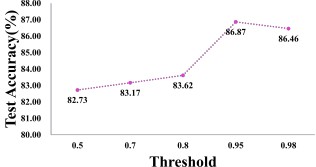

Figure 9: Test accuracy on FashionMNIST, under varying sizes of the synthetic dataset.

Figure 10: Test accuracy on FashionMNIST, under varying $\lambda_2$ in the local update.

Figure 11: Test accuracy on FashionMNIST, under varying thresholds.

