# OpenReview forum: "Federated Learning with GAN-based Data Synthesis for Non-IID Clients"
_ICLR.cc/2022/Conference — ICLR 2022 Submitted_

### Official Review · Reviewer_UMsE · 2021-10-30

**Correctness:** 3
**Technical Novelty And Significance:** 2
**Empirical Novelty And Significance:** 2
**Recommendation:** 5
**Confidence:** 4

**Main Review:**

Strengths:
1. The studied problem is important. Non-IID data is a key challenge in federated learning.
2. From the experiments, the improvement of the proposed approach is significant.

Weaknesses:
1. Some important details about the algorithm are missing. What is the size of the local synthetic data per client? Is it related to the size of the local data?
2. The GAN seems to be an important part of the algorithm. The paper directly adopts WGAN without convincing explanation. The authors can add experiments to show the effect of the GAN architectures.
3. The communication overhead may be significantly large than FedAvg since the synthetic data need to be communicated each round. The authors should theoretically or empirically analyze the communication cost.
4. In Algorithm 1, why the client needs to update the synthesis dataset? The dataset has already been updated by the server.
5. The authors claim that the local data including both the real and the synthetic data approach an IID distribution. However, there is no convincing explanation to support it.
6. The experiments are weak. MNIST and Fashion-MNIST are two simple tasks, which are not challenging in federated learning. For the results on CIFAR-10, the differentially private version does not show superior performance compared with FedAvg.
7. There is no experimental study on the hyperparameters including $\lambda$ and the threshold. The authors should add them to understand the proposed algorithm more clearly.
8. What is the non-IID data setting in the experiments of different privacy budget?
9. Typo: the second paragraph of “Effect of privacy budget”: Table 5 -> Table 6.


**Summary Of The Paper:**

The paper proposes a new federated learning algorithm called SDA-FL, which utilizes GANs to generate synthetic data for federated training on non-IID data. Specifically, each client pretrains a GAN to generate synthetic data and send them to the server. In each round, the server sends the global model and the synthetic data to the clients. Then, the clients update the label of the synthetic data and use both the local data and synthetic data to update the local model. The local models are sent to the server, which further averages the models and uses the averaged model to label the synthetic data. Experiments show that SDA-FL significantly outperforms the other federated learning approaches on non-IID data. The paper also studies the influence of different privacy budgets on the performance when applying differentially private GANs.

**Summary Of The Review:**

The paper proposes a new federated learning algorithm on non-IID data. Although the proposed algorithm is interesting, many details in the algorithm and experiments are missing. I believe the paper needs to be significantly improved from both the theoretical and empirical perspectives.

---

> ### Author Response · Authors · 2021-11-22
> **Response to Reviewer UMsE's Review**
>
> We thank the reviewer for the detailed reading of the paper and valuable feedback. We address the issues pointed out by the reviewer as follows:
>
> 1\. Some important details about the algorithm are missing.
>
> Thank you for your comment. In SDA-FL, every client uploads 4000 samples to the server to constitute the synthetic dataset together.
>
> In the experiment, the size of the local synthetic data is independent of the size of the local real data.
> However, the size of the local synthetic data influences the model performance.  To demonstrate how the size of the synthetic dataset affects the performance, we conducted an ablation study on FashionMNIST in Appendix C of our revision, and the results showed that compared with uploading 4000 samples per client, there is only a 1.4% drop in the test accuracy with uploading 200 samples per client.
>
> 2\. The choice of WGAN
>
> Thank you for your comment. The selection of the GAN architecture is to ensure stable training, as well as its capability to generate synthetic data that is of sufficiently high quality for improving the federated training performance. The GAN models adopted in previous studies (e.g., CGAN [3], DCGAN [4]) are very unstable and difficult to train because the Jensen-Shannon divergence loss function frequently leads to the gradient vanishing and mode collapse problems [1] [2]. In comparison, WGAN can solve these problems and generate synthetic data stably [5]. More explanations and discussion are presented in Section 3.
>
> [1] https://arxiv.org/pdf/1701.04862.pdf
>
> [2] https://arxiv.org/pdf/1701.07875.pdf
>
> [3] https://arxiv.org/pdf/1411.1784.pdf
>
> [4] https://arxiv.org/pdf/1511.06434.pdf
>
> [5] https://arxiv.org/pdf/1701.07875.pdf
>
> 3\. The communication overhead may be significantly large than FedAvg
>
> 4\. Why the client needs to update the synthesis dataset?
>
> 5\. The local data including both the real and the synthetic data approach an IID distribution.
>
> Response to problems 3-5:
>
> Thanks for your comments. In our framework, all the clients send their generated data to the server and construct a synthetic dataset, which is a one-shot process before the federated training. Then, the synthetic dataset is distributed to all the clients. As each generative model can produce the representative samples from the local dataset, local training based on the local samples as well as the synthetic samples can alleviate the non-IID problem.
>
> In each communication round, the server will update the pseudo labels of the synthetic dataset and send the updated labels as well as the global model to each client. The interplaying labeling process can gradually improve the confidence level of the pseudo labels, as the global model gradually converges with the increasing of iterations. Empirical results shown in Fig. 6 illustrate that keeping updating pseudo labels for the synthetic data improves the model performance.
>
> Note that as the size of the updated pseudo labels is much smaller than the number of parameters, the extra communication cost is neglectable in each round. Besides, the communication overhead of sharing the synthetic samples is proportional to the size of the synthetic dataset. To investigate the influence of the size of the synthetic dataset on the model performance, we conducted an ablation study on the FashionMNIST classification task (Appendix C). As illustrated in Fig. 9, the proposed method has only a small (1.5%) drop in the accuracy when the size decreases from 50,000 to 2,000. This indicates that the proposed method is not sensitive to the size of the synthetic data, and the communication overhead is tolerable.
>
> 6\. The experiments are weak.
>
> We have included the Cifar-10 and SVHN datasets to evaluate the proposed method in Section 5. Besides, for differential privacy experiments on Cifar-10, with 2 classes of data for each client, SDA-FL reaches 67.17% in the test accuracy and it outperforms all the baselines including Fedmix and Fedprox. Please refer to task 3 in section 5 of our revision.
>
> 7\. There is no experimental study on the hyperparameters including $\lambda$ and the threshold.
>
> Thanks for your suggestions about the ablation studies of the hyper-parameters. We have included ablation studies about the size of the synthetic dataset, the computational cost, the number of samples for training the GAN, $\lambda_2$, and the threshold. Please refer to the details on task 5 of Section 5 and Appendix C of our revision.
>
> 8\. What is the non-IID data setting in the experiments of different privacy budgets?
>
> Thank you for this comment. We conducted the experiments of different privacy budgets on MNIST, FashionMNIST, and CIfar-10 with each client keeping 2 classes of data.
>
> 9\. Typo: the second paragraph of “Effect of privacy budget”: Table 5 $\rightarrow$ Table 6.
>
> Thank you for pointing out this typo. We have revised it accordingly.

---

> > ### Comment · Reviewer_UMsE · 2021-11-30
> > **Thanks for your response**
> >
> > Some of my concerns have been addressed. Thus, I have raised my score to 5. However, I still cannot recommend acceptance due to the significant changes in the Experiment section after revision. I cannot check all the details of the new Experiment section. For problem 2, WGAN and the mentioned studies are published before 2017. I think there should be more state-of-the-art studies on GANs recently.

---

> > > ### Author Response · Authors · 2021-12-01
> > > **Thanks for your further response**
> > >
> > >
> > > Thanks for your valuable time and your constructive comments
> > >
> > > In the revision, we mainly supplement some experiments to further demonstrate the effectiveness of the SDA-FL.
> > >
> > > In task 1, we included the Cifar-10 and SVHN datasets in the comparisons with other traditional algorithms, and the results still show a large margin.
> > >
> > > In task 2, we added the Cifar-10 dataset in the semi-supervised learning experiments, which also illustrate the benefits of SDA-FL in terms of the convergence rate and performance.
> > >
> > > In task 3, we included the FID scores to further clarify how the privacy budgets affect the quality of the GAN-based data and the overall performance. The results illustrate that SDA-FL still maintains a significant performance even under the strict privacy protection.
> > >
> > > In task 4, we added the experiments to demonstrate the effectiveness of the "pseudo labeling process" and the "server update", which are the two improvements over the traditional FL. The results show that these two procedures in SDA-FL help to improve the overall performance significantly.
> > >
> > > In task 5, we added the experiments to show the influence factors to the quality of the GAN-based data, including the computational cost and the number of samples used to train the GANs. The results show that SDA-FL still outperforms other baselines under the acceptable computational cost and the small number of samples for the GAN training.
> > >
> > > Besides, the experiments on Covid-19 dataset and other ablation studies are shown in the appendix, demonstrating the generalizability and the feasibility of the SDA-FL.
> > >
> > > For problem 2, we agree with the reviewer that there have been other better GANs architectures recently, but it is important to note that the SDA-FL framework has demonstrated its effectiveness with the application of the WGAN, and we believe that SDA-FL can achieve the better performance with more efficient GANs. Thank you once more for your suggestions for improving the GANs.

---

### Official Review · Reviewer_Ena1 · 2021-11-01

**Correctness:** 3
**Technical Novelty And Significance:** 2
**Empirical Novelty And Significance:** 2
**Recommendation:** 3
**Confidence:** 4

**Main Review:**

The main strengths are:
1. Utilized the generative power of GAN to simulate the distribution of local data, without directly sharing it with peer and central server. Also used differential privacy, the two methods both maintained the local data privacy;
2. Utilized pseudo labeling to further maintain the generated dataset.

The main weaknesses are:
1. The privacy issue itself. Consider the extreme case, where each local client only keeps a single class of data. In this case, the first round locally pre-trained GAN-generated data uploaded to the central server will mostly contain the data which has the same label, which immediately break the privacy of the local data;
2. Lack of citations. The differential private GAN is not an original idea. This idea has been broadly explored in the GAN community recently, thus proper citations are needed. Also, more citations of federated learning algorithms focusing on non-iid setting are also missing, for example, [1][2][3];
3. The algorithms are not described clearly in the algorithm boxes. ServerUpdate is described as ''is the same as the
LocalUpdate execpt the data'', which needs more clarification;
4. Lack of details of experiments and code. Most of the details of the experiments, for example, training hyperparameters, network details are missing. Since no code is provided, it's kind of hard for one to repeat the experiments and confirm the effectiveness of the algorithm;
5. Lack of experiments. More experiments should be conducted, for example for the first task "Federated supervised learning,"  different ways of partitioning the data should be considered, rather than the extreme case alone;
6. Lack of theoretical analysis;
7. Quality of GAN-generated images. This can be a minor issue, but the quality of the GAN-generated images provided in the paper is kind of low, not sure whether it's because of the noisy gradients;
8. Another minor issue is why different choices of differential budget are only provided in the CIFAR-10 task?


[1] Tackling the objective inconsistency problem in heterogeneous federated optimization. J Wang, Q Liu, H Liang, G Joshi, HV Poor - arXiv preprint arXiv:2007.07481, 2020

[2] H. Wang, Z. Kaplan, D. Niu and B. Li, "Optimizing Federated Learning on Non-IID Data with Reinforcement Learning," IEEE INFOCOM 2020 - IEEE Conference on Computer Communications, 2020, pp. 1698-1707, doi: 10.1109/INFOCOM41043.2020.9155494.

[3] On the convergence of fedavg on non-iid data. X Li, K Huang, W Yang, S Wang, Z Zhang - arXiv preprint arXiv:1907.02189, 2019

**Summary Of The Paper:**

This paper focused on a classical federated learning setting where data was non-iid partitioned in different local servers, and came up with a method named SDA-FL, which combined GAN-generated data, differential privacy to both address the non-iid problem and keep local data privacy. The main contributions are:
1. Utilized differential private GAN-generated data to solve the non-iid and local data privacy problems;
2. Designed a label updating mechanism to increase the model performance;
3. Tested the algorithm on CIFAR-10, MNIST, fashion-MNIST to confirm the performance of the algorithm.

**Summary Of The Review:**

This paper focused on a challenging problem -- non-iid in federated learning and tried to combine methods from different areas to solve it, which is interesting itself, however, this paper mainly suffers from the following aspects:
1. Technical correctness. The privacy is not well maintained;
1. Lack of novelty. The proposed algorithm is more like a rough combination of several existing methods, with little innovative thoughts or theoretical explanations;
2. Design of experiments. More experiments should be carefully designed and conducted;
3. Missing proper citations.

---

> ### Author Response · Authors · 2021-11-22
> **Response to Reviewer Ena1's Review**
>
> 6\. Lack of theoretical analysis
>
> Thanks for this comment. The objective of this paper is to develop an efficient federated learning framework to resolve the non-IID problem in FL by generating a global shared synthetic dataset.
>
> Most previous works [3] [4] focus on the local update and aim to keep the local models close to each other. However, we intend to address the non-IID problem by GAN-based data augmentation. GANs have been widely used in federated learning to relieve the non-IID and data scarcity problems [1] [2].  The authors of [1] adopt the CGAN to generate images with pseudo labels, but it is unstable and frequently suffers from the gradients vanishing and mode collapse issues.  The method proposed in [2] suffers from the pseudo labeling problems and cannot generate confident labels, while in our work, the SDA-FL method can make good use of the synthetic data and the real data by constituting the synthetic dataset and providing high-confidence pseudo labels.  In addition, the effectiveness of the proposed methods is validated by extensive experimental results on various benchmark datasets, and our method achieves better performance than the state-of-the-art methods while providing privacy protection. With the above said, it will be interesting to develop theoretical results for the proposed methods, which will be pursued in our future studies.
>
> [1]https://arxiv.org/pdf/2110.07136.pdf
>
> [2]https://arxiv.org/pdf/2104.12581.pdf
>
> [3]https://arxiv.org/pdf/1812.06127.pdf
>
> [4]https://arxiv.org/pdf/1910.06378.pdf
>
> 7\. Quality of GAN-generated images.
>
> We agree with the reviewer that the quantity of the synthetic samples is sensitive to the noisy gradients caused by the differential privacy mechanism. We showed the trade-off between privacy levels and the quality of the synthetic data (task 3 in Section 5), where the image quality is quantified by the FID score [5]. We also conducted experiments about how the computational cost affects the quality of the synthetic data (task 5). With the decreasing of the privacy levels and the computational cost, the FID score increases significantly, which affects the performance. The GAN-generated images provided in the paper are also degraded due to these two factors, but it is more important to note that SDA-FL still outperforms other baselines with such low-quality synthetic data, which implies that the quality of the synthetic data would not heavily influence the model performance.
>
> [5] https://arxiv.org/pdf/1706.08500.pdf
>
> 8\. Another minor issue is why different choices of differential budget are only provided in the CIFAR-10 task?
>
> Thank you for your suggestion. We have added experiment results in  MNIST, FashionMNIST, and Cifar-10 datasets to investigate the privacy-performance tradeoff. More details about the settings and results are shown in Section 5 of our revision.

---

> ### Author Response · Authors · 2021-11-22
> **Response to Reviewer Ena1's Review**
>
> We thank the reviewer for the detailed reading of the paper and valuable feedback. We address the issues pointed out by the reviewer as follows:
>
> 1\. Label privacy issue
>
> Thanks for your comment. First of all, in this extreme case, we cannot rule out the possibility of the label leakage you mentioned.  However, compared with the acceptable label leakage, SDA-FL is able to achieve its maximum effectiveness in solving the non-IID issue in this extreme case. The results in Table 1 of our revision show that other methods intrinsically suffer from the non-IID issue on MNIST, FashionMNIST, Cifar-10 and SVHN when each client only keeps one class of data, while SDA-FL is able to achieve a significantly better performance.
>
> In addition, we assert that SDA-FL is still effective at protecting the privacy of the synthetic data because it adopts the widely used  ($\epsilon$,$\delta$)-differential privacy as the metric of the privacy level. The privacy experiments (task 3 in section 5) on MNIST, FashionMNIST, and Cifar-10 also demonstrate that SDA-FL still outperforms other methods even in a strict privacy protection level.
>
> 2\. Lack of citations
>
> Thank you for your suggestion. We have improved the manuscript and added the references accordingly.
>
> 3\. The description of the ServerUpdate
>
> Thank you for pointing out this issue. Both the server and clients perform stochastic gradient descent based on the same loss function (6). We have added more details and revised Algorithm 2 in the manuscript.
>
>  4\. Lack of details of experiments
>
> Thank you for this comment. We have provided more details about the hyperparameters in Appendix A of the revised manuscript.
>
> 5\. Lack of experiments
>
> Thanks for your reminds about the experiments. We have added Cifar-10 and SVHN into the federated supervised learning. We also test every dataset in three different non-IID scenarios. All the results are shown in Table 1 of the revised manuscript, which demonstrates the performance gain of the proposed method.
>
> Besides, we have conducted experiments to investigate the trade-off between privacy level, computational cost, and performance to provide more insights on this work and make the presentation of empirical results more comprehensive. Please refer to Section 5 of our revision for more details.

---

> > ### Comment · Reviewer_Ena1 · 2021-11-30
> > **Thanks for your response**
> >
> > I thank the authors for the clarifications. My concerns regarding the experiment part are addressed. However, I am still worrying the proposed algorithm breaks the privacy restriction, also the extreme case issue has not not been well explained. Another issue is that the training time spent on training a GAN, the real-world data is much more complicated than MNIST and CIFAR10, thus the training time can be huge, compared to the time spent on the FL stage. For these reasons, I'll keep my initial score.

---

> > > ### Author Response · Authors · 2021-12-01
> > > **Thanks for your further response**
> > >
> > > Thank you for the comment.
> > >
> > > For your concern about the privacy leakage, firstly in SDA-FL, all the clients only send their generated data rather than the generators to the server, which avoids the privacy issues introduced by them. To protect the clients' information in the generated data, we adopt the differential privacy mechanism. The experimental results in task 3 of the manuscript show that SDA-FL can outperform the baselines with strict privacy protection. Particularly, the generated Cifar-10 images with the privacy budget $\epsilon=200$ are shown in Fig.4(a). These images are extremely blurry and unrealistic compared with the real data, which implies that the GAN models would not lead to severe privacy leakage.
> > >
> > > Considering possible label leakage in the extreme case, where each local client only keeps a single class of data, we can employ a proxy server as the middle layer between the server and all the clients to solve this problem, which achieves anonymity [1] of clients in model aggregation. Although the server may recover the label information, it does not know where the data come from.
> > >
> > > Besides, we agree with the reviewer that the computational overhead of training a GAN model is high, and we have conducted an ablation study (task 5 in Section 5.2) to investigate the tradeoff between the GAN training complexity and the performance. However, it is important to note that our SDA-FL method could work perfectly in the cross-silo scenarios, where each client has sufficient computational resources for the GAN training. Recently, medical institutions have used GANs to address the non-IID issue in the medical data [2] [3]. In particular, we have included an experiment on the real Covid-19 dataset [4] in Appendix B, and the results demonstrate that SDA-FL is more effective than other baselines as well as the algorithm proposed by [3].
> > >
> > > We hope that the response can alleviate your concern.
> > >
> > > [1] https://ieeexplore.ieee.org/document/9325934
> > >
> > > [2] https://arxiv.org/pdf/2110.07136.pdf
> > >
> > > [3] https://arxiv.org/pdf/2104.12581.pdf
> > >
> > > [4] https://arxiv.org/pdf/2003.13145.pdf

---

### Official Review · Reviewer_t36G · 2021-11-02

**Correctness:** 3
**Technical Novelty And Significance:** 2
**Empirical Novelty And Significance:** 2
**Recommendation:** 5
**Confidence:** 4

**Main Review:**

Strength-

* The proposed idea is clear and well-presented: The idea to use partially-private synthetic data to resolve the non-IID issue in federated learning is interesting and clearly presented.
*Experimental result shows the improvement in performance: The authors compare their method with other federated learning methods and demonstrate the improvement in performance from the sharing of synthetic data. The trade-offs between privacy level and the quality of synthetic data are also presented.

Weakness-

* The overhead of training local GANs in each client can be huge: The proposed scheme achieves the performance gain from sharing of synthetic data, which models the private local data with some privacy protection. The performance gain highly depends on the quality of synthetic data, but in real scenarios, it could be unrealistic to train GANs in every client using their local data to generate high-quality synthetic data.
* More empirical support would be necessary: the proposed scheme highly depends on the quality of pseudo-labels as well as the synthetic data. There are many performance measures to quantify the quality of synthetic data and pseudo labels, i.e., FID and Inception Score for synthetic data and label accuracy for the pseudo labels, which can be measured by training a classifier with the synthetic data. Quantifying such quality and examining trade-offs between privacy level, computational cost, quality of synthetic data and the overall performance will provide better insights on this work and make the presentation of empirical results more comprehensive.


**Summary Of The Paper:**

This paper proposes a new framework for federated learning to resolve the non-IID issue by sharing differentially private synthetic data. Each client pretrains a local GAN to generate synthetic data and upload the data to the parameter server. To effectively use the synthetic data, the server performs pseudo labeling and shares this information with the clients along with the global model parameters, to let the local data, including both the real and synthetic data, approach an IID distribution. The interplay between model training and synthetic data updating improves the convergence of the local models and resolves the non-IID issue.

**Summary Of The Review:**

The proposed idea of using synthetic data, generated by local GANs of each client, and pseudo labelling at PS seems to be effective to resolve the non-IID issue of federated learning. However, the overhead of training local GANs in each client can be huge, which makes this approach not very practical. The overhead should be clearly addressed to compare this work with other baselines. The current experimental results only measure the overall test accuracy, but since the trade-offs between quality of synthetic data, pseudo-labelling, and overall performance are main factors in the proposed scheme, presenting such trade-offs in more detailed manner would be helpful to support the idea.

---

> ### Author Response · Authors · 2021-11-22
> **Response to Reviewer t36G's Review**
>
> We thank the reviewer for the detailed reading of the paper and valuable feedback. We address the issues pointed out by the reviewer as follows:
>
> 1\. The overhead of training local GANs in each client
>
> To address the reviewer's concern about the quality of the synthetic data, we conducted an ablation study (task 5) to investigate how the overhead of training local GANs affects the performance. Particularly, we trained the generators with 10, 30, 50, 100, and 200 rounds (each round with 90 steps) on the FashionMNIST classification task, and each client keeps 2 classes of data.  The results in Fig.7 show that there is a tradeoff between the training overhead (i.e., the number of training rounds) and the model performance. Although SDA-FL suffers performance degradation when decreasing the training rounds, it still outperforms Fedmix (82.41%) and Fedprox (75.76%).
>
> Besides, our proposed method could fit perfectly in the cross-silo scenarios, where each client has sufficient computational resources for GAN training. To further demonstrate the feasibility of the proposed framework, we conduct an experiment on the medical field, which performs pneumonia classification on a Covid-19 dataset [3]. We compare SDA-FL with the baselines as well as the method proposed in [1].  The experimental results in Table 6 illustrate that SDA-FL is superior to other methods by a remarkable margin.  Please refer to the experiment details in Appendix B of our revised manuscript.
>
> [1]https://arxiv.org/pdf/2110.07136.pdf
>
> [2]https://arxiv.org/pdf/2104.12581.pdf
>
> [3]https://arxiv.org/pdf/2003.13145.pdf
>
> 2\. More empirical support about quantifying the quality and examining trade-offs between privacy level, computational cost, quality of synthetic data,  and the overall performance.
>
> Thank you for your suggestion. We have selected FID as a metric to measure the quality of the synthetic samples and conducted experiments to investigate the trade-off between the privacy level, computational cost, quality of synthetic data, and overall performance.
>
> (a) The trade-off between privacy budget and performance
>
> We have conducted the ablation study (task 3 in Section 5) to illustrate the privacy performance tradeoff.  In this task, we evaluated the performance of SDA-FL on MNIST, FashionMNIST, and Cifar-10 under different privacy budgets. The results are shown in Fig.3. Compared with the free-protection GANs ($\epsilon=\infty$), a strict privacy level leads to a significant increase of the FID score and slight degradation of the overall performance. However, SDA-FL still outperforms Fedmix and Fedprox in MNIST, FashionMNIST, and Cifar-10 classification tasks.
>
> (b) The trade-off between computational cost and performance
>
> To investigate the influence of the computational cost on the model performance, we evaluated SDA-FL on the FashionMNIST dataset given different numbers of rounds for training the GANs (task 5 of Section 5). The results in Fig.7 show that low computational cost does affect the quality of the synthetic data and the performance, but it is important to note that SDA-FL demonstrates its effectiveness in data augmentation and outperforms other methods even with ten rounds for training the generators.
>
> We hope these additional experiments can help the reviewer better understand SDA-FL.

---

### Official Review · Reviewer_ex8V · 2021-11-02

**Correctness:** 3
**Technical Novelty And Significance:** 2
**Empirical Novelty And Significance:** 2
**Recommendation:** 6
**Confidence:** 4

**Main Review:**

Strengths:

1. This paper addresses important non-IID and privacy problems in federated learning. It is a straightforward idea to share synthetic data instead of real data to avoid privacy leak. DP-GAN also has theoretically guarantee of privacy preserving. The framework seems to be promising.
2. The performance of proposed framework is good.
3. The effect of GAN-synthetic data is evaluated by a comparison with closely related algorithms. There is an improvement comapred to NaiveMix and FedMix.
4. The sensitivity to privacy budget is evaluated.
5. Each additional component seems not so novel, but it is a reasonable and potentially useful combination.
6. Clear delivering of a complicated system.

Weaknesses:

1. Equation (4) seems to be wrong. If the algorithm use vanilla mixup, the input of classifier $$f(\cdot; \boldsymbol{w}_i)$$ should be the mixed data $$\bar{\boldsymbol{X}}_i$$, rather than $$(1 - \lambda_1) \boldsymbol{X}_i$$ or $$\lambda_1 \hat{\boldsymbol{X}}_i$$.  $$\lambda_1$$ should be $$1 - \lambda_1$$ to align with equation (3), and vise versa. Same mistake in Algorithm 2. The author should correct them or explain.

2. Inadequate evaluation of labelling process. It is not verified whether the interplaying labeling process is better than some naive method, e.g.,

   - Each client trains a local classifier to assign pseudo label to its synthesis data, once and for all.
   - Each client uses conditional GAN or similar techniques to generate images directly given label.

   The author should carefully design ablation study to evaluate this.

3. Potential advantages introduced by ServerUpdate. An additional ServerUpdate step is introduced to the framework. Therefore, SDA-FL may take more gradient descent steps than baselines (FedAvg, FedProx, FedMix, etc.) in a communication round. People may be skeptical whether it is the additional gradient steps make the model converge faster. The author should make sure that same number of gradient steps in taken in each round.

4. Lack of details and unclear hyperparameters in experiments. Number of clients, number of class per client for Cifar-10, $$\alpha, \lambda_2, E, B$$, hyperparameters for baselines... are not provided. More experiments details should be provided in main text, supplementary materals, or released code.

5. Error bar is not provided. People may ask whether figure results are from only one run. It will be much better to provided result with standard deviation or confidence interval, together with the number of independent runs.

Other concerns and questions:

1. In SDA-FL, each participant train a GAN before training the classifier, introducing the following requirements:

   - Each client must have sufficient data to train GAN.
   - Each client must have adequate computing resources. (GAN often requires more resources. )
   - In each communication round, clients are selected from *ones that uploaded their GANs*. This may implicitly requires a sustainable or high participation rate of client.

   These requirements may limit the scope of SDA-FL's application. In cross-device settings, it's hard for personal devices (like mobile phones) to meet the above requirements. For me it seems like SDA-FL can only be applied to cross-silo settings. Please comment.

2. DP-GAN guarantee privacy preserving during data synthesis. The training process of classifier is not protected and might leak personal information. This is an orthogonal problem but the author may mention this in the text.

**Summary Of The Paper:**

Federated learning suffers from non-IIDness across clients. A line of previous works address the non-IID problem by data sharing which violates the privacy requirement. In this paper, the author propose SDA-FL. Compare to the most basic FedAvg, the additional components includes

- *Image synthesis with DP-GAN*. Before training classifier, each client trains a differentially private GAN to generate synthetic data, and upload these data to PS for future data-sharing.
- *Synthetic image labeling*. Similar to self-training, PS use local models to assign pseudo-labels to unlabeled data. These labels are updated during training, i.e., interplay between model training and synthetic dataset updating.
- *Mixup*. For each clients, private data and shared synthetic data are mixed to alleviate non-IIDness.
- *ServerUpdate*. Data sharing also makes it possible for the server to conduct gradient descent.

This paper also empirically evaluate their framework under supervised and semi-supervised learning settings. Moreover, this paper study the sensitivity to privacy budget and effect of synthetic data.

**Summary Of The Review:**

This paper addresses important non-IID and privacy problems in federated learning, and proposes a frameworks with some components are evaluated. However, components of interplaying labeling and ServerUpdate are not well-evaluated in this paper. Moreover, the details of experiments are inadequate.

---

> ### Author Response · Authors · 2021-11-22
> **Response to Reviewer ex8V's Review**
>
> 6\. The concern about the sufficient data, computation resources and the high participation rate of clients.
>
> Thank you for your comment. We address the raised issues as follows:
>
> (a) 'Each client must have sufficient data to train GAN' and 'Each client must have  adequate computing resources'.
>
> We agree with the reviewer that a large-scale dataset and powerful computational resources are necessary to train a high-quality GAN and produce high-realistic samples. However, the objective of the proposed method is to improve the model performance and alleviate the non-IID problem, rather than generating perfect synthetic samples.
> We have found that low-quality samples (with the confidence level larger than a given threshold) can effectively improve the global model performance. In the revised manuscript, we have conducted ablation studies (task 5 in Section 5) to investigate the impact of the number of training samples and the overhead of training local GAN on the performance.
>
> The experimental results are depicted in Fig. 7 and Fig. 8, which show that insufficient training rounds and limited training samples lead to low-quality synthetic data and degraded performance. The generator only gets an FID score of 217.81 (23.17 for real data) and an accuracy of 83.76\% when training the generators with 30 rounds. In addition, as shown in Fig. 8, under the fixed 30-round computational cost, using fewer samples to train the generators reduces the quality of the synthetic data and thus affects the performance. Nonetheless, despite the low computational cost (30 rounds) and a relatively small number of samples (1000 samples) used to train the generators individually, our framework still outperforms other baselines (82.41\% in Fedmix), which demonstrates the robustness of the proposed method.
>
> Besides, our proposed method could fit perfectly in the cross-silo scenarios, where each client has sufficient computational resources for GAN training. To further demonstrate the practical applicability of the proposed framework, we conduct an experiment on the healthcare field, which performs pneumonia classification on a Covid-19 dataset [5].
> We compare SDA-FL with the baselines as well as a recent method proposed in [6].
> The experimental results in Table 6 illustrate that SDA-FL is superior to other methods by a remarkable margin. Please refer to the experiment details in Appendix B of our revised manuscript.
>
> (b)  'In each communication round, clients are selected from ones that uploaded their GANs'.
>
> In our framework, the clients only upload their synthetic data before the training process, rather than uploading their GANs, which helps to avoid the privacy leakage introduced by the generators.  After the synthetic dataset is established, the server only updates the pseudo labels of the synthetic samples in each communication round.  Thus, SDA-FL does not require a high participation rate of clients.
>
> [3]https://arxiv.org/pdf/2110.07136.pdf
>
> [4]https://arxiv.org/pdf/2104.12581.pdf
>
> [5]https://arxiv.org/pdf/2003.13145.pdf
>
> [6]https://arxiv.org/pdf/2104.12581.pdf
>
> 7\. The privacy protection of  local classifiers
>
> We agree with the reviewer that the classifier might leak personal information, and this is an open problem in federated learning and out of the scope of our study. We have mentioned this point in Section 3 of the manuscript.

---

> ### Author Response · Authors · 2021-11-22
> **Response to Reviewer ex8V's Review**
>
> We thank the reviewer for the detailed reading of our paper and the valuable feedback. The responses to the comments are provided as follows
>
> 1\. Equation (4) seems to be wrong.
>
> Thank you for pointing out this typo. We have revised it accordingly.
>
> 2\. Inadequate evaluation of labeling process.
>
> Thank you for the constructive comment.
>
> We have conducted an ablation study in section 5.2 to evaluate the effectiveness of the labeling process.  In particular, we tested the performance of the proposed SDA-FL method by setting the updating round to 2, 10, and 20.
> The results in Fig. 5 of our revision show that the test accuracy drops with the decreasing of the update rounds, which indicates that the interplaying labeling process outperforms the naive methods.
>
> For the conditional GAN (CGAN), we agree with the reviewer that CGAN can simultaneously generate the synthetic data with pseudo labels.
> However, the CGAN model frequently suffers from gradient vanishing and mode collapse problems as mentioned in [1] and [2], particularly with the differential privacy noise, which will degrade the quality of the synthetic samples. Simply using the low-quality synthetic samples from CGAN may hurt the federated training performance. In contrast, the selected WGAN-GP method can overcome the aforementioned problems, because the Wasserstein divergence is continuous and differentiable compared with the Jensen-Shannon divergence used in the CGAN, which has also been mentioned in [2].
> Besides, we introduced a labeling mechanism that adopts the maximum class probability (MCP) as the metric to measure the confidence level of each sample and only generates pseudo labels to the high-quality samples if its MCP value is larger than a threshold.
> Therefore, WGAN-GP combined with our labeling process is better than the CGAN method.\
> [1] https://arxiv.org/pdf/1701.04862.pdf \
> [2] https://arxiv.org/pdf/1701.07875.pdf
>
> 3\. Keep the number of gradient steps the same in each round:
>
> Thank you for your comment. We agree that the extra gradient descent steps in the SeverUpdate may lead to unfair comparison. Therefore, we have modified the experimental setting to maintain that the total gradient steps (i.e., the summation of the local steps and the server steps) of SDA-FL are the same as other baselines in each communication round.
> The corresponding results are shown in Fig. 2, which still demonstrates the efficient convergence of the proposed method.
>
> 4\. Lack of details and unclear hyperparameters in experiments.
>
> Thank you for this comment. We have provided more details about the hyperparameters in Appendix A of the revised manuscript.
>
> 5\. Error bar is not provided.
>
> Thank you for your suggestion. We have shown the results with error bars in Fig.3 and Fig.4.

---

> > ### Comment · Reviewer_ex8V · 2021-11-28
> > **Thanks for your response**
> >
> > Question 1, 3, 4, 5 are well-answered. I appreciate releasing the detailed hyper-parameters, which helps assessing the reproducibility.
> >
> > For question 2, I was interested in whether it is possible to generate both image and labels once and for all. I can think of two possible ways: local model labeling and conditional GAN (CGAN), both not considered. For CGAN, I understand that the SDA-FL framework can help selecting high quality samples. However, I disagree that CGAN degrades the quality of synthetic samples: conditionality and divergence function are two orthogonal improvement based on original GAN. They can be easily combined (here is a blog).
> >
> > http://blog.richardweiss.org/2017/07/21/conditional-wasserstein-gan.html
> >
> > Meanwhile, I agree that Figure 6 (instead of 5) shows that updating pseudo-labels helps model training. Thus the effect of labeling process is partially evaluated.

---

> > > ### Author Response · Authors · 2021-11-29
> > > **Thanks so much for your further response**
> > >
> > >
> > > In SDA-FL, the server utilizes the local classifiers to update the pseudo labels for the corresponding synthetic data generated from the same client. Because the local classifier and the corresponding synthetic data come from the same client, we believe the effect of this process is the same as "local model labeling."
> > >
> > > Furthermore, thanks again for your constructive suggestions about the conditional-WGAN. We will compare the performance of the WGAN with that of the conditional-WGAN to improve the generalizability of the SDA-FL in future work.

---

### Author Response · Authors · 2021-11-22
**Summary of Changes in the Revision**

We thank all the reviewers for their thorough and insightful feedback. We revised our paper following the comments and made the changes below.



**Major Updates**

1\. We  have  included  Cifar-10  and  SVHN  datasets  on  federated  supervised  learning(task 1 in section 5) to further demonstrate the effectiveness of SDA-FL.

2\. We have included Cifar-10 for federated semi-supervised learning (task 2 in section5).

3\. We  have  used  FID  to  quantify  the  quality  of  the  GAN-based  data  and  show  the trade-off between the privacy level and quality as well as performance (task 3 in section5).

4\. We have added the experiments to illustrate the effectiveness of the pseudo labeling process  and  server  update,  which  are  the  contributions  of  SDA-FL  compared  with  the traditional federated learning (task 4 in section 5).

5\. We  have  added  the  ablation  studies  about  computational  cost  and  the  number  of samples for training the GANs (task 5 in section 5).

6\. We have supplemented all the experimental details in Appendix A.

7\. We have added a comparison on the Covid-19 dataset in Appendix B.

8\. We have added the ablation studies about the size of the dataset,λ2and the threshold in Appendix C.

**Minor updates**

1\. We have added the appropriate references as suggested.

2\. We have classified the server update process more clear in Algorithm 2.

3\. We have added a further discussion about why we adopt WGAN instead of the traditional GANs in section 3.

4\. We have corrected all minor typos and errors in text and formulas.



We believe that our paper has become stronger and more well-founded with this revision. We'd like to thank the reviews once more for their constructive feedback.

---

### Decision · Program_Chairs · 2022-01-20

**Decision:**

Reject

**Comment:**

This paper proposes an idea to address the non-IID issue that is well-known in federated learning. After the discussion with the authors, there are still some concerns remained about the proposed approach. First and foremost, the training of local GAN at each client can be demanding computationally and statistically, which limits the practicality of their approach. Secondly, there has been other work that aims to study the non-IID issues in federated learning, as suggested by the reviewers. The authors should consider citing some of the work in this literature and compare the prior approaches with their GAN-based approach. Thirdly, there is a lack of a formal statement about the privacy guarantee in this paper. In particular, it seems that the privacy guarantee would only make sense in the cross-silo setting, in which each client has many users' data. If each client corresponds to a single user, it does not make sense to train a local GAN. The authors should consider elaborating on the privacy guarantee in the next revision.